# The Modernization of Oriental Music Therapy: Five-Element Music Therapy Combined with Artificial Intelligence

**DOI:** 10.3390/healthcare12030411

**Published:** 2024-02-05

**Authors:** Chan-Young Kwon, Hyunsu Kim, Sung-Hee Kim

**Affiliations:** 1Department of Oriental Neuropsychiatry, College of Korean Medicine, Dong-eui University, Busan 47227, Republic of Korea; 2Department of Automotive Engineering, Dong-eui University, Busan 47340, Republic of Korea; hkim7@deu.ac.kr; 3Department of Industrial ICT Engineering, Dong-eui University, Busan 47340, Republic of Korea; sh.kim@deu.ac.kr

**Keywords:** ICT, art therapy, five-element music therapy, non-pharmacological treatment, EATM

## Abstract

In recent years, music has been regarded as a promising non-pharmacological intervention for a number of physical and mental conditions. Five-elements music therapy—based on the five-element theory—is a unique non-pharmacological therapy of East Asian traditional medicine. It has the potential to effectively provide individualized music therapy to individuals with illness. However, one limitation of this music therapy is that the classification of the five elements and its application is mainly based on subjective judgment. The development of artificial intelligence (AI) has enabled the acoustic analysis of multi-factor sound sources. This can develop five-element music therapy. Here, we discussed the challenges proposed by the future combination of five-element music therapy and AI. Further, we hypothesized that AI may promote its use in the medical field.

## 1. Introduction

Music has been a part of human history for tens of thousands of years. It has diversified through human genetic and cultural evolution to have four purposes, including dance, personal or communal entertainment, communication, and ritual [1,2]. Moreover, music is a promising intervention for improving human behavior, emotions, and health because it contributes to human empathy, cooperative behavior, altruistic behavior, and positive emotions [1]. The concept of music therapy for healing was established in ancient times; however, its scientific foundations, including brain science, have been added since the 20th century [3]. Music therapy is defined as “the clinical and evidence-based use of music interventions to accomplish individualized goals within a therapeutic relationship by a credentialed professional who has completed an approved music therapy program” by the American Music Therapy Association [4]. Recently, evidence-based practice of music therapy has received increasing attention in this field [5].

East Asian traditional medicine (EATM), including traditional Chinese medicine, Korean medicine, and Kampo medicine, has been developed based on two core concepts: (1) a close relationship between humans and the cosmos (i.e., environments); and (2) a dynamic balance for maintaining integrity between them [6]. In this context, EATM has developed the five-element theory, Wood, Fire, Earth, Metal, and Water, to explain and relate human biological and cosmic phenomena [6]. This theory proposes that the five elements explain all phenomena in the universe and play a role in connecting humans (i.e., Liver, Heart, Spleen, Lung, and Kidney) and the universe [6,7]. One of the characteristics of this theory is the specific relationship (i.e., mutual interaction and rotation) between the five elements [7]. The mutual interaction between the elements includes the mutual nourishment cycle and mutual restraint cycle [7], and EATM has developed unique treatment theories using this relationship. One of the EATM treatments is five-element music therapy.

Under the system of EATM, music or sound can be classified by the five-element theory into five tones: Jue, Zhi, Gong, Shang, and Yu. This classification is based on the classics of EATM, the Yellow Emperor’s Internal Classic [8]. Jue, Zhi, Gong, Shang, and Yu are the basic scales of ancient Chinese music, which correspond to the seven-tone scales of Western music, do, le, mi, sol, and la, respectively [9]. This classification has been used for both diagnosis through auscultation and treatment through music therapy [8]. The five tones are associated with Wood, Fire, Earth, Metal, and Water (i.e., the five elements), respectively [9]. Similarly, human emotions are classified into anger, joy, thinking, sorrow/worry, and fear/fright and are associated with Wood, Fire, Earth, Metal, and Water, respectively. Therefore, the five music tones are considered to be associated with human emotions [9]. Taken together, the five-element theory conceptualizes that the five tones can be used to control human emotions directly or indirectly via five-element music therapy. For example, anger corresponds to Wood, while Metal suppresses Wood; therefore, listening to the sound of Metal (i.e., Shang) is a therapeutic approach to control excessive anger. Likewise, sorrow corresponds to Metal, while Fire suppresses Metal; therefore, listening to the sound of Fire (i.e., Zhi) is a therapeutic approach to control excessive sorrow (Figure 1). Based on this mutual interaction, the psychopathologies that could be controlled using five-element music therapy include rumination, depression, anxiety, anger, and mania (Table 1).

In this study, we performed a mini literature review and bibliometric analysis to investigate the feasibility of modernizing music therapy. Moreover, the possibility of combining it with information and communication technology (ICT) and its challenges were discussed.

## 2. Mini Literature Review: Clinical Benefits of Five-Element Music Therapy

One strength of five-element music therapy includes its use as a dynamic and individualized therapy. It is derived from the five-element theory, which emphasizes the dynamic balance between humans and environments. For example, the five-element theory can interpret the dynamics of various human emotional states and create individualized music therapy to reinforce or suppress the corresponding emotions [9]. The importance of individualized and specialized music therapy is consistent with the current emphasis on conventional music therapy [10]. Secondly, five-element music therapy is different from conventional music therapy because its diagnostic system is shared with other EATM treatments, including moxibustion, acupuncture, herbal medicine, and acupressure. Therefore, any imbalance among the five elements is the basis for the application of acupuncture or herbal medicine at the same time as the music therapy. This suggests that a combination of five-element music therapy and other EATM therapies is highly likely. Thirdly, this music therapy originated in EATM, which values the relationship between the body and mind [6]. Therefore, this treatment can affect human emotions and physical health by acting on the Liver, Heart, Spleen, Lung, and Kidney [9].

To investigate the clinical benefits of five-element music therapy, we conducted a mini literature review of a systematic review on five-element music therapy. MEDLINE via PubMed was searched by using the following search strategy, with no limit to the search range: (five-elements OR five elements) AND music AND systematic. The search data date was 10 November 2023, and 18 articles were found. Among them, four systematic reviews [11,12,13,14] were included. It was found that just as the evidence-based practice of music therapy is considered important in Western society [5], the effectiveness of the five-element music therapy has also been studied from the perspective of evidence-based medicine. Recent systematic reviews [11,12,13,14] on the clinical effectiveness of five-element music therapy are summarized in Table 2. Notably, Zhu et al. [11] concluded that five-element music therapy is likely to be better than ordinary music for antenatal depression. Notable original studies included in these reviews are summarized in Appendix A.

However, the underlying therapeutic mechanisms of five-element music therapy have been insufficiently studied in humans. Instead, some non-clinical experiments have sought to investigate the therapeutic mechanisms of music therapy. For example, Yuan found that in a rat model of post-stroke depression, when acupuncture and the five-element music therapy were used together, norepinephrine content in the prefrontal cortex and serotonin content in the hippocampus were significantly increased compared to the group that was only administered acupuncture [15]. Cheng et al. found that the antidepressant effect of the five-element music therapy in a rat model of depression was associated with a significant decrease in the expression of NLRP3 and IL-1β in the hippocampus [16]. These results suggest the possibility that music therapy may work through regulating brain neurotransmitters and reducing neuroinflammation, but its therapeutic mechanism needs to be further elucidated.

## 3. Bibliometric Analysis: The Trends of Studies of Five-Element Music Therapy

Although five-element music therapy originated in the East, a recent multi-country study [17] found that it has the potential to work effectively in Western populations as well. This randomized controlled trial conducted by Liao et al. [17] compared five-element music therapy with Western-based music therapy in 35 individuals including Canadian and Chinese participants. As a result, regardless of the cultural context, both types of music therapy were significantly effective for subjective stress, and only five-element music therapy was associated with significant improvements in anxiety and depression [17]. In the same context, recent studies on music therapy are increasingly accessible not only from local databases in China but also from international databases such as MEDLINE (via PubMed). The search date was 10 November 2023, and the search strategy was as follows, with no limit to the search range: (five-elements OR five elements) AND music (Figure 2).

To analyze research topics related to five-element music therapy, the Web of Science Core Collection was also searched, with the same search strategy. As results, a total of 44 documents were found. The bibliographic data were analyzed using a VOS viewer. Among the 207 keywords in these studies, 33 keywords occurred in multiple studies. The most frequent keywords in this field included depression, five-element music, acupuncture, and anxiety (Table 3).

The network of keywords was visualized as an occurrence network map, with overlay visualizations. According to the map, the recent keywords among the studies of five-element music therapy included the coronavirus disease of 2019, traditional Chinese medicine, exercise, symptoms, and anxiety (Figure 3).

## 4. Challenges and Opportunities of Five-Element Music Therapy with Artificial Intelligence

Among the challenges of five-element music therapy, the most important one is that the classification of the five tones is dependent on subjective experience. There have been attempts to understand the five tones as musical components, such as by using frequencies (i.e., Hz) [18,19] and scales (i.e., Do, Re, Mi, Sol, and La) [20]; however, no proven objective method has enabled the classification of sound sources to date. The classification of the five tones has not yet been defined or agreed upon in an objective way including frequency or wavelength. Moreover, music affects human experience through multidimensional interaction; therefore, it has been difficult to objectively define the properties of the five-element music therapy. This has been the same problem not only in five-element music therapy, but also in other treatments including herbal medicine, acupuncture, and qigong, which share the same EATM theory. Importantly, some researchers have emphasized the potential of artificial intelligence (AI) to solve the challenges of treatments based on the EATM theory [21]. Also, the development of AI provides an opportunity to analyze the multifactorial elements of music itself and the complex elements of the human experience of music. The use of ICT is a way to overcome the limitations of five-element music therapy and to strengthen its benefits. We are already aware of successful adaptations of music therapy using AI (Table 4). However, there have been no attempts yet to combine five-element music therapy and AI. Recently, AI chatbots available to the public, such as Chat GPT, have provided a response to five-element music therapy, including a simple example of sheet music (Table 5).

Based on this basic knowledge, we propose the following challenges for boosting five-element music therapy with AI. First, the typical sound source of five-element music therapy could be analyzed to define the five tones. For example, a time-frequency analysis can be performed using a fast Fourier transform to understand the acoustic characteristics of a typical five-element sound source. Moreover, an index for the acoustic definition of the five tones could be developed using signal processing techniques. For example, a wavelet transform and cepstrum could be used to detect signal/sound changes over a short period and specify the frequency repetition, respectively. Second, an algorithm to analyze the properties of the five elements could be developed. This process should take into account that the typical sound source of five-element music therapy is mainly based on oriental instrumental music. One recent study attempted to extract the characteristic parameters of traditional Chinese instrumental music with the mel frequency cepstral coefficient [25]. This case can also be considered for the characterization of five-element music therapy. Third, the five-element properties of other sound sources could be analyzed. Specifically, sound sources for five-element music therapy should be investigated by analyzing the properties of the five tones using a defined algorithm for non-five-element sound sources. Such an algorithm has the potential to be used in individualized music therapy with sound sources from widely accessible platforms, such as YouTube. Fourth, a new sound source could be created using AI with defined five-element properties. For example, a recent research team described a style transfer framework for providing personalized music therapy called SleepGAN [26]. They analyzed sleep-promoting music and other music based on six indices, including ‘articulation’, ‘energy’, ‘spectrum’, ‘rhythm’, ‘bass and treble’, and ‘noise- or tone-like’, to transform existing music into sleep-promoting music by weighting 34 defined musical features [26]. These technologies can be used to develop and apply new sound sources for five-element music therapy to promote individualized music therapy and achieve freedom from copyright issues. Fifth, real-time customized five-element music therapy could be provided based on the individual’s dynamic emotional response. Therefore, this can provide real-time therapy based on the individual’s subjective emotional state, or bio-signs suggesting their emotional state, including voice, electroencephalography, electrocardiography, heart rate, and heart rate variability. By solving these challenges, five-element music therapy will be able to provide not only the curative benefits of ordinary music-based interventions [27,28], but also unique benefits based on the five-element theory (Figure 4).

## 5. Conclusions

Five-elements music therapy has been used as a non-pharmacological therapy in EATM. Recent advances in ICT enable the standardization and objectification of this therapy. This is because ICT is thought to be helpful in objectifying the subjective classification of the five tones, which is the most important challenge of five-element music therapy. This potential development enables the objective manipulation of five-element music therapy, indicating that it can be individualized for each patient. Recent clinical trials have reported that five-element music therapy has a promising therapeutic effect on mental and physical disorders [29,30,31]. The use of ICT, combined with the five-element music therapy, is thought to create an opportunity to promote its use by addressing the limitations related to the delivery of this music therapy.

## Figures and Tables

**Figure 1 healthcare-12-00411-f001:**
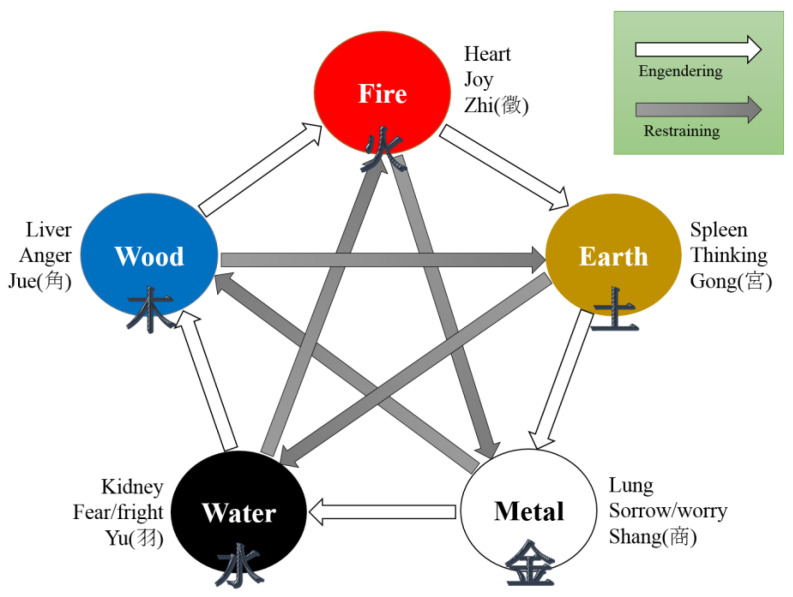
The five-element theory and five tones.

**Figure 2 healthcare-12-00411-f002:**
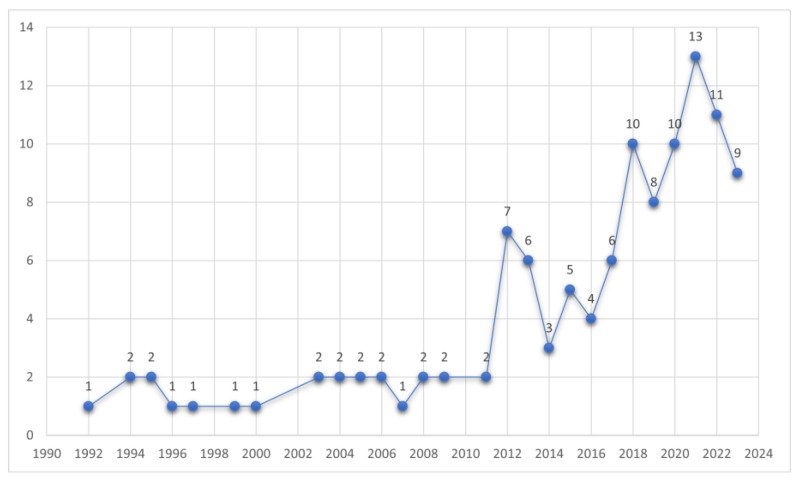
Number of five-element music therapy studies retrieved from MEDLINE (via PubMed). Note: The *x*-axis represents the year of publication, and the *y*-axis represents the number of studies.

**Figure 3 healthcare-12-00411-f003:**
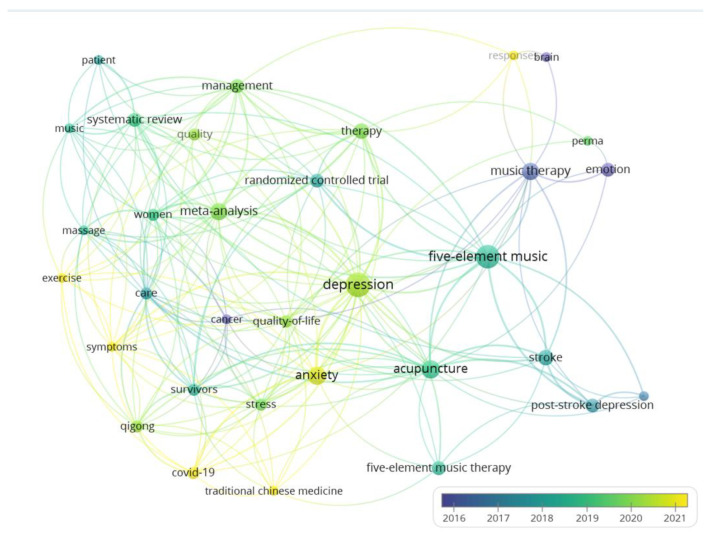
The keyword occurrence map of five-element music therapy studies retrieved from the Web of Science Core Collection. Note: The size of the node indicates its frequency of occurrence.

**Figure 4 healthcare-12-00411-f004:**
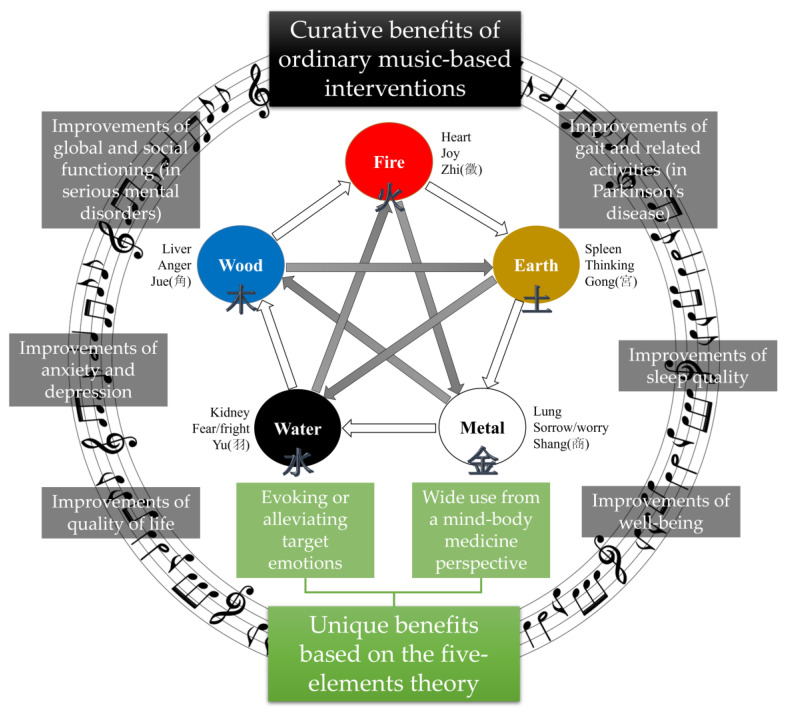
Curative benefits of ordinary music-based interventions and unique benefits based on the five-element theory.

**Table 1 healthcare-12-00411-t001:** Concept of emotional regulation using five-element music therapy.

The Five Elements	The Five Organs	The Five Tones	Emotions Evoked	Emotions Inhibited	Associated Psychopathology
Wood	Liver	Jue	Joy	Worry	Rumination
Fire	Heart	Zhi	Worry	Sorrow	Depression
Earth	Spleen	Gong	Sorrow	Fear/fright	Anxiety
Metal	Lung	Shang	Fear/fright	Anger	Anger
Water	Kidney	Yu	Anger	Joy	Mania

**Table 2 healthcare-12-00411-t002:** Clinical effectiveness of five-element music therapy investigated in systematic reviews.

1st Author (Year)	Database (Search Period)	Clinical Conditions	Intervention	Number of Included Studies	Main Findings
Zhu (2021) [11]	Cochrane Library, PubMed, CNKI, WanFang, WoS, Embase (inception to April 2021)	Antenatal depression	Music, massage, yoga, and exercise	24 RCTs (3 RCTs for five-element music therapy)	“*This research found low to very low evidence that yoga, exercise, music and massage could reduce antenatal depression. Among them, music may be the most effective intervention, and integrated yoga other than simple yoga would improve prenatal depression. The effect of five-element music may be better than ordinary music.*”
Yang (2021) [12]	PubMed, Cochrane Library, Embase, WoS, CNKI, WanFang, CBM and VIP (inception to December 2020)	Cancer	Five-elements music therapy	22 RCTs	“*Five-element music therapy had a positive effect on depression, quality of life, sleep quality, and Karnofsky performance score in cancer patients, while did not show a positive effect on anxiety.*”
Wu (2020) [13]	Web of Science, PubMed, Embase, Cochrane Library, CNKI, WanFang and VIP (inception to January 2020)	Perinatal mental health and labor pain	Five-elements music therapy	13 RCTs	“*Five-element music may be efficacious in improving perinatal women’ depression, anxiety, labor pain, labor duration, the hemorrhage 2 h after parturition and the Serum dynorphin expression level.*”
Yang (2019) [14]	MEDLINE, PubMed, CNDTR, CSGTR, CENTRAL, WoS, Embase, CINAHL, PsycINFO, Science Direct, LILACS, AMED, CBM, CNKI, WanFang, VIP, ITPLS, ClinicalTrials.gov, CCTR and BIOSIS Previews (inception to May 2018)	Poststroke aphasia	Five-elements music therapy	6 RCTs	“*Five-element music might moderately improve language rehabilitation in individuals with poststroke aphasia.*”

Abbreviations: AMED, Allied and Alternative Medicine; CBM, Chinese Biomedical Literature Database; CCTR, Chinese Clinical Trial Register; CENTRAL, Cochrane Central Register of Controlled Trials; CINAHL, Cumulative Index to Nursing and Allied Health Literature; CNDTR, Cochrane Neuromuscular Disease Group Trials Register for randomized trials; CNKI, China National Knowledge Infrastructure; CSGTR, Cochrane Stroke Group Trials Register; ITPLS, Index to Taiwan Periodical Literature System; LILACS, Latin American and Caribbean Health Sciences Literature; RCT, randomized controlled clinical trial; VIP, China Science and Technology Journal Database; WoS, Web of Science.

**Table 3 healthcare-12-00411-t003:** Frequent keywords among the studies on five-element music therapy.

Rank	Keyword	Occurrences	Total Link Strength
1	depression	12	96
2	five-element music	11	61
3	acupuncture	7	46
3	anxiety	7	37
5	meta-analysis	6	59
5	music therapy	6	37
7	stroke	5	30
7	therapy	5	56

**Table 4 healthcare-12-00411-t004:** Cases of successful introduction of artificial intelligence in music therapy.

1st Author (Year)	Intervention	Clinical Condition	AI Introduced	Improvements in Music Therapy
Russo (2023) [22]	Music-based digital therapeutic	BPSD	Two AI systems including biological music information retrieval and affective music recommendation system	“*Extracting insights about the emotional states that are likely to be evoked by pieces of music and generating playlists that are personalized to each user.*”
Modran (2023) [23]	Music-based smartphone application	Not specified	Machine learning	“*Development of machine learning model that could predict whether a specific song has therapeutic effects on a specific person.*”
Williams (2020) [24]	Generative music technology	Mental health conditions	Machine learning algorithm	“*Producing infinite soundtracks in sympathy with a listener’s bio-signals, and in a biofeedback loop.*”

Abbreviations: AI, artificial intelligence; BPSD, behavioral and psychological symptoms of dementia.

**Table 5 healthcare-12-00411-t005:** Chat GPT-3.5’s response to a query about five-element music therapy.

Question	“Explain the Characteristics of Music that Corresponds to the Music of Element Water in the Five-Elements Music Therapy, and Produce Sheet Music.”
Chat GPT-3.5’s response	*The concept of associating musical elements with the five traditional elements (earth, water, fire, air, and ether) in music therapy is a creative and subjective approach. While there is no universally agreed-upon set of characteristics for music corresponding to the element of water, certain qualities are often associated with water in various cultural and artistic contexts. These characteristics might include:* *1. Fluidity and Flow:* *Water is often linked with smooth, flowing movements. In musical terms, this could translate to legato phrasing, flowing melodies, and a sense of continuous motion.* *2. Calmness and Serenity:* *Water is often associated with calmness and serenity. Music representing water might include gentle dynamics, slow tempos, and a peaceful overall mood.* *3. Reflectiveness:* *Water can reflect light and surroundings. Similarly, music associated with water might have reflective qualities, perhaps using techniques like echo effects or musical themes that mirror each other.* *4. Depth and Mystery:* *Water is deep and mysterious. Music reflecting water might include elements of depth, possibly through the use of lower registers, sustained notes, or harmonic complexity.* *5. Rhythmic Pulsation:* *Water often has a rhythmic quality, whether it’s the rhythmic lapping of waves or the steady flow of a river. Musical compositions might incorporate a rhythmic pulse reminiscent of water’s natural movements.* *6. Harmonic Resonance:* *Water has a unique way of resonating with different frequencies. Music associated with water might explore harmonic resonance, perhaps through the use of open chords or specific harmonic progressions.* *7. Use of Water Instruments:* *Incorporating instruments with a watery sound, such as the flute, harp, or certain percussion instruments, can enhance the association with the element of water.* *Here’s a simple example of sheet music that embodies some of these characteristics. Keep in mind that this is a basic representation, and the interpretation of water in music can vary widely:* *Water Sonata* *Composer: [Your Name]* *Tempo: Adagio* *Key: C Major* *[Piano Intro]* *| C G/B | Am G | F C/E | Dm7 G7 |* *[Main Theme]* *| C G/B | Am G | F C/E | Dm7 G7 |* *| C G/B | Am G | F C/E | Dm7 G7 |* *| Em Am | Dm7 G7 | C E | Am G |* *[Reflective Section]* *| F G7 | C E | Am G | F G7 |* *| C G/B | Am G | F C/E | Dm7 G7 |* *[Coda]* *| Em Am | Dm7 G7 | C ||* *Feel free to modify the sheet music to better fit your interpretation of water in music therapy. Adjustments in dynamics, tempo, and instrumentation can further enhance the representation of water’s characteristics in the piece.*

Abbreviations: AI, artificial intelligence; BPSD, behavioral and psychological symptoms of dementia. Note: The example of sheet music is a response from Chat GPT, and no manipulation has been made by the authors. The alphabet and numbers in the sheet music represent the chords, and ‘|’ is a line to separate the measures.

## Data Availability

Data are contained within the article.

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
