# Peer review of "The Modernization of Oriental Music Therapy: Five-Element Music Therapy Combined with Artificial Intelligence"

_healthcare, 2024, doi:10.3390/healthcare12030411_

Round 1

Reviewer 1 Report

Comments and Suggestions for Authors

This paper gives a detailed explanation of five-elements music therapy in a clinical setting and the potential for further understanding and utilisation through AI.

This paper was structured well and easy to read. I would make recommendation to accept this paper with minor revisions to consider the following couple of points:

1. In Table 1, fourth column, the scale (Do, Re, Mi etc) is listed. Judging by this table, it would seem that there is a definitive pitch matching the other five elements and concepts of the five-elements music therapy. However, later on you put forward that one of the uses for AI could be to further determine what would be the pitches/frequencies of the five-elements. Therefore it seems slightly confusing that they are presented in the table, as it is something not yet definite. I would make a suggestion of removing them from the table, however, this may be at your discretion. 

2. Page 6, line 14- please indicate what 'one' you are referring to in the sentences " the most important one..."

Author Response

  • Response to Comments from Reviewer 1

Overall comment:

This paper gives a detailed explanation of five-elements music therapy in a clinical setting and the potential for further understanding and utilisation through AI.

This paper was structured well and easy to read. I would make recommendation to accept this paper with minor revisions to consider the following couple of points:

Response:              

Thank you very much for taking your valuable time to review this manuscript. We have no doubt that the reviewer’s comments will help to further improve the quality of this manuscript.

Comment 1:

  1. In Table 1, fourth column, the scale (Do, Re, Mi etc) is listed. Judging by this table, it would seem that there is a definitive pitch matching the other five elements and concepts of the five-elements music therapy. However, later on you put forward that one of the uses for AI could be to further determine what would be the pitches/frequencies of the five-elements. Therefore it seems slightly confusing that they are presented in the table, as it is something not yet definite. I would make a suggestion of removing them from the table, however, this may be at your discretion.

Response:              

Thank you for the comment, and we agree with it. Therefore, to eliminate unnecessary misunderstandings, the scale has been removed from Table 1 as advised.

Comment 2:

  1. Page 6, line 14- please indicate what 'one' you are referring to in the sentences " the most important one..."

Response:

Thank you for the comment. We modified the sentence as follows to reflect the reviewer's comments:

“Among the challenges of five-elements music therapy, the most important one is that the classification of the five tones is dependent on subjective experience.”

(Please refer page 6, red words)

Reviewer 2 Report

Comments and Suggestions for Authors

Thank you for your interesting manuscript. I found it to be informative and stimulating.

The following are my thoughts/comments/suggestions/questions:

1. Review for English – overall, it is very good; however, please review for the inclusion of small words such as “a” or “the”.

2. Page 1, line 29 – research into brain science is definitely newer; however, I think the scientific research into music therapy began in the 20th century. Please double check this.

3. Page 2 – great figure. It is very helpful to see the five-elements theory and five tones illustrated in this manner.

4. Page 2 – Table 1 is also very helpful. You might want to review it though and make sure that “worry” is capitalized both times it is listed as this is the only word in the table that was not capitalized.

5. Page 5, line 128 – “33 keywords were occurrence more than a study” – I think you might mean “33 keywords occurred in multiple studies”. Is this what you mean? If not, please clarify as the sentence is confusing.

6. Page 5 – I love the keyword occurrence map! This is very interesting and provides a unique way to look at this information.

7. Page 6 – it was interesting to see Chat GPT’s response to the question about the characteristics of music that correspond to water. This was a very helpful example.

8. Page 7 – I’m confused by the “sheet music”. It is technically a listing of chords, but is there a melody? How long is each chord supposed to be held? Does it represent “water” no matter the length of time spent on each chord? Is this supposed to be something that someone can use in therapy and have each client adapt it? Did one of the authors compose this – or was it composed by AI?

9. Page 8, line 197 – could you please provide more information on how ICT can standardize and objectify five-elements music therapy? And, is it meant to be standardized, or should it be personalized for each patient?

10. Page 8, line 200 – could you please provide more information on how ICT and five-elements music therapy could be used more in the medical field? This seems like a over-arching statement that isn’t necessarily supported by the rest of the document.

11. Page 8, reference #4 – this isn’t the correct reference for this source. This definition of music therapy is from 2005 and can be found on the AMTA website.

Comments on the Quality of English Language

Overall it is very good. There just need to be some minor edits.

Author Response

  • Response to Comments from Reviewer 2

Overall comment:

Thank you for your interesting manuscript. I found it to be informative and stimulating.

The following are my thoughts/comments/suggestions/questions:

Response:              

Thank you very much for taking your valuable time to review this manuscript. We have no doubt that the reviewer’s comments will help to further improve the quality of this manuscript.

Comment 1:

  1. Review for English – overall, it is very good; however, please review for the inclusion of small words such as “a” or “the”.

Response:              

Thank you for the comment. We have reviewed and corrected the errors (especially those involving determiners) the reviewer commented throughout the manuscript. All changes within the manuscript are marked in red.

Comment 2:

  1. Page 1, line 29 – research into brain science is definitely newer; however, I think the scientific research into music therapy began in the 20th century. Please double check this.

Response:              

Thank you for the comment, and we agree with it. Therefore, based on the reviewer's comments, we modified the existing sentence as follows:

“The concept of music therapy for healing was established in ancient times; however, its scientific foundations, including brain science have been added since the 20th century [3].”

(Please refer page 1, red words)

Comment 3:

  1. Page 2 – great figure. It is very helpful to see the five-elements theory and five tones illustrated in this manner.

Response:              

Thank you for your comment. We hope that this figure will help readers understand five-elements theories and five tones more easily.

Comment 4:

  1. Page 2 – Table 1 is also very helpful. You might want to review it though and make sure that “worry” is capitalized both times it is listed as this is the only word in the table that was not capitalized.

Response:              

Thank you for the comment. Thank you for finding that error. We capitalized the ‘W’ in worry to unify it with other emotions.

(Please refer Table 1, red words)

Comment 5:

  1. Page 5, line 128 – “33 keywords were occurrence more than a study” – I think you might mean “33 keywords occurred in multiple studies”. Is this what you mean? If not, please clarify as the sentence is confusing.

Response:              

Thank you for the comment. The comment the reviewer pointed out is exactly what we intended. Therefore, we modified the sentence as follows:

“Among the 207 keywords in these studies, 33 keywords occurred in multiple studies.”

(Please refer page 5, red words)

Comment 6:

  1. Page 5 – I love the keyword occurrence map! This is very interesting and provides a unique way to look at this information.

Response:              

Thank you for the comment. We hope that this figure will provide readers with an easy visual understanding of the current state of research related to five-element music.

Comment 7:

  1. Page 6 – it was interesting to see Chat GPT’s response to the question about the characteristics of music that correspond to water. This was a very helpful example.

Response:              

Thank you for the comment. Thank you for your interest in this example. We were also surprised that Chat GPT gave a more detailed response than we expected regarding the five-elements music therapy. And we hope that this will be conveyed to the readers as well.

Comment 8:

  1. Page 7 – I’m confused by the “sheet music”. It is technically a listing of chords, but is there a melody? How long is each chord supposed to be held? Does it represent “water” no matter the length of time spent on each chord? Is this supposed to be something that someone can use in therapy and have each client adapt it? Did one of the authors compose this – or was it composed by AI?

Response:              

Thank you for the comment. The sheet music shown in Table 5 is the response from Chat GPT, and none of the authors have modified it. Chat GPT does not provide detailed information about this code, but provides a simple example listed in Table 5. We've added clarification to clarify this point.

Abbreviations. AI, artificial intelligence; BPSD, behavioral and psychological symptoms of dementia. Note. The example of sheet music is a response from Chat GPT, and no manipulation has been made by its authors.”

(Please refer page 7, red words)

Comment 9:

  1. Page 8, line 197 – could you please provide more information on how ICT can standardize and objectify five-elements music therapy? And, is it meant to be standardized, or should it be personalized for each patient?

Response:              

Thank you for the comment. We have made the following corrections to clarify the pointed sentences:

“Five-elements music therapy has been used as a non-pharmacological therapy used in EATM. Recent advances in ICT enable the standardization and objectification of this therapy. This is because ICT is thought to be helpful in objectifying the subjective classification of the five tones, which is the most important challenge of five-elements music therapy. This potential development enables the objective manipulation of five-element music therapy, indicating that it can be individualized for each patient. Recent clinical trials have reported that five-elements music therapy has a promising therapeutic effect on mental and physical disorders [25-27].”

(Please refer page 8, red words)

Comment 10:

  1. Page 8, line 200 – could you please provide more information on how ICT and five-elements music therapy could be used more in the medical field? This seems like a over-arching statement that isn’t necessarily supported by the rest of the document.

Response:              

Thank you for the comment. In response to the reviewer's comments, we have revised the final sentence of the conclusion as follows. In particular, we aimed to maintain a consistent flow of content throughout this manuscript.

“The use of ICT, combined with the five-elements music therapy, is thought to create an opportunity to promote its use by addressing the limitations related to the delivery of this music therapy.”

(Please refer page 8, red words)

Comment 11:

  1. Page 8, reference #4 – this isn’t the correct reference for this source. This definition of music therapy is from 2005 and can be found on the AMTA website.

Response:              

Thank you for the comment. The reference was changed as recommended by the reviewer.

“American Music Therapy Association. What is Music Therapy? Available online: http://www.musictherapy.org/about/musictherapy/ (accessed on Jan 16, 2024).”

(Please refer page 9, red words)
